# Understanding and Improving the Training of Data-Efficient GANs

## Abstract

Recently, many studies have highlighted that training data-efficient generative adversarial networks (DE-GANs) suffers from the overfitting of the discriminator ($D$) problem. However, how this issue theoretically influences the training of the generator ($G$) remains unclear. In this paper, we unveil a novel insight in DE-GANs training, regarding the significance of useful gradients of $G$. Notably, the useful gradients of $G$ are those computed on generated samples closer to the real data distribution, which helps the generator to better learn the real data distribution. As the overfitting degree of $D$ increases, the gradients of the $G$ become progressively less useful, which hinders DE-GANs training. Based on this novel insight, we propose a simple yet effective general training strategy for DE-GANs to provide more useful gradients for $G$ by selecting fake samples to update $G$ during training, namely adaptive Top-k (ATop-k). Concretely, only the Top-k high-score fake samples are used to update $G$, where the value of k is controlled by the overfitting degree of $D$ adaptively. Extensive experiments on several datasets demonstrate that ATop-k can effectively improve the training of DE-GANs and achieve better performance with different DE-GANs.

## 1 Introduction

Generative Adversarial Networks (GANs) (Goodfellow et al., 2014) have been remarkably successful (Karras et al., 2018; Miyato & Koyama, 2018; Karras et al., 2019; 2020b; Yu et al., 2021; Jiang et al., 2021b) in generating high-quality data when working with large datasets. However, collecting and cleaning these large datasets can be expensive, time-consuming, and even infeasible. Therefore, data-efficient generative adversarial networks (DE-GANs) (Li et al., 2022b; Zhao et al., 2020) have received great attention.

Recently, more and more studies (Zhao et al., 2020; Karras et al., 2020a; Yang et al., 2021; 2022a) have focused on DE-GANs and highlighted that DE-GANs are prone to the overfitting of the discriminator ($D$) issue during training. However, it is still unclear how this issue influences the training of the generator ($G$) theoretically. To answer this, we present a novel insight regarding the significance of useful gradients of $G$ in DE-GANs training. As discussed in Sinha et al. (2020) and Wu et al. (2019), the useful gradients of $G$ are those computed on generated samples closer to the real data distribution (Sinha et al., 2020; Wu et al., 2019), which leads to the generator learning a better model of the data distribution. When the overfitting degree (Karras et al., 2020a) of $D$ increases, i.e., $D$ becomes more and more confident in distinguishing both real and fake data, the overlapping parts, i.e., the overlap between the supports of real and fake data distributions, become smaller and smaller (Karras et al., 2020a). As shown in Figure 1, the progressive disappearance of overlapping parts indicates that the generated samples are further and further away from the real data distribution (Karras et al., 2020a), which leads to the gradients of the generator ($G$) becoming less and less useful. Finally, as pointed out in Arjovsky & Bottou (2017) and Goodfellow et al. (2014), the overlapping parts disappear or can be ignored when $D$ becomes overly confident, i.e., $D$ reaches optimality. This results in the gradient instability or vanishing problem of the generator ($G$). Without loss of generality, we analyze this novel insight theoretically using two widely-used DE-GAN backbones with different loss functions, i.e., StyleGAN2 (Karras et al., 2020b) with non-saturating loss and FastGAN (Liu et al., 2021) with hinge loss (refer to §3.1).

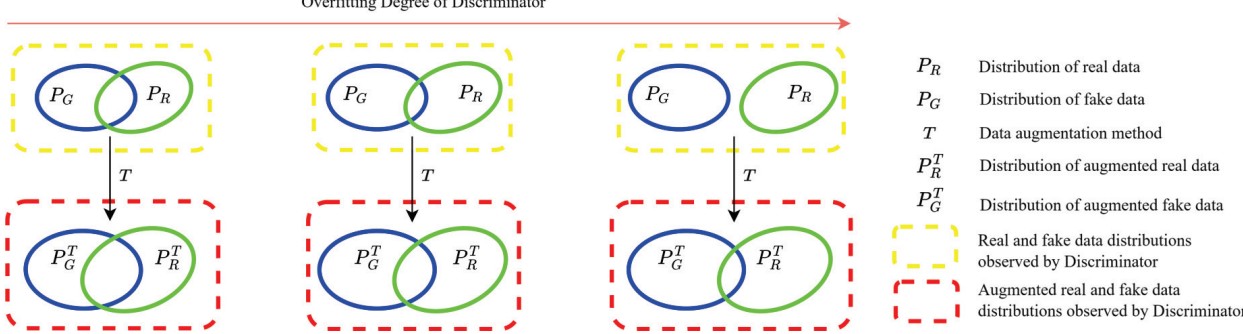

Figure 1: An illustration of the novel insight regarding the significance of useful gradients of $G$ in DE-GANs training. According to the useful gradients of $G$ as defined in Sinha et al. (2020) and Wu et al. (2019), we demonstrate the overlapping parts during training, which can effectively show how close the fake data distribution is to the real data distribution, to illustrate the novel insight. **Top**: The real and fake data distributions during training. The overlapping parts exist in the initial training state. As the overfitting degree of $D$ increases, the overlapping parts become smaller and smaller, and the gradients of $G$ become less and less useful. When $D$ becomes overly confident, i.e., $D$ reaches optimality, the overlapping parts disappear or can be ignored, causing the gradient instability or vanishing problem of $G$ (see §3.1). **Bottom**: The augmented real and fake data distributions during training. Data augmentation can enlarge the overlapping parts in the initial training state by augmenting both real and fake data distributions, which makes more generated samples closer to the real data distribution, thus providing more useful gradients for $G$ during training and leading to better performance.

Existing approaches (Zhao et al., 2020; Karras et al., 2020a; Liu et al., 2021; Fang et al., 2022; Tseng et al., 2021; Sauer et al., 2021; Kumari et al., 2022; Wang et al., 2023b) employ various strategies on $D$ such as data augmentation, model regularization and pre-trained models to improve the DE-GANs training. Among these studies, data augmentation (DA) is the widely-used approach for all these methods to improve the training of DE-GANs. To demonstrate the generalization ability of the proposed novel insight, we theoretically connect it with DA and highlight that DA in DE-GANs can prevent $D$ from becoming overly confident, thereby maintaining the overlapping parts, which indirectly leads to more useful gradients for $G$ during training. Specifically, DA in DE-GANs can enlarge the overlapping parts in the initial training state by augmenting both real and fake data distributions, which makes more generated samples closer to the real data distribution, thus resulting in better performance.

Driven by the proposed insight, we improve the DE-GANs training task from a novel perspective: Directly providing more useful gradients for $G$ by selecting fake samples to update $G$ during training. This selection process is guided by the critic scores (Azadi et al., 2018; Sinha et al., 2020), i.e., the discriminator outputs, on the fake samples. To this end, we propose a general training strategy called adaptive Top-k (ATop-k) for DE-GANs. Specifically, only the Top-k high-score fake samples, i.e., the generated samples that are the closest to the real data distribution, are used to update the generator ($G$), where the value of k is controlled by the overfitting degree of $D$ adaptively. As a result, ATop-k enables $G$ to directly obtain more useful gradients during training, thus yielding better performance.

To sum up, the main contributions are as follows: (1) We unveil a novel insight regarding the significance of useful gradients of $G$ in DE-GANs training. Specifically, as the overfitting degree of $D$ increases, the overlapping parts become progressively smaller, resulting in the gradients of the $G$ becoming progressively less useful; (2) We propose a general training strategy called adaptive Top-k (ATop-k) for DE-GANs. ATop-k effectively enables $G$ to obtain more useful gradients during training, thereby achieving better performance; (3) Experiments on low-shot (Zhao et al., 2020) and FFHQ (Karras et al., 2020b) datasets with different DE-GANs demonstrate that ATop-k can improve performance across various DE-GANs and deliver state-of-the-art performance.

## 2 Related Work

### 2.1 Generative Adversarial Networks (GANs)

Generative adversarial networks (GANs) (Goodfellow et al., 2014) are a form of generative model that consists of two players: a generator ($G$) and a discriminator ($D$). Concretely, $G$ generates synthetic data with some given noise while $D$ distinguishes whether the data is from the generator's output or real data. GANs are well-known to suffer from instability of training, resulting in poor quality and lack of diversity of generated samples. To stabilize the training of GANs as well as increase the quality and diversity of generated samples, various methods have been proposed, focusing on more sophisticated advanced architectures (Miyato & Koyama, 2018; Miyato et al., 2018; Zhang et al., 2019; Radford et al., 2016; Brock et al., 2019; Yang et al., 2022b), more stable loss functions (Arjovsky et al., 2017; Gulrajani et al., 2017; Mao et al., 2017; Salimans et al., 2016; Guo et al., 2020), and better training strategies (Karras et al., 2018; Denton et al., 2015; Liu et al., 2020; Zhang et al., 2017; Li et al., 2022a) to achieve better results.

### 2.2 Data-Efficient GANs (DE-GANs)

Recently, investigating how to train GANs when sufficient data is unavailable has become increasingly popular and meaningful. Nearly all studies (Zhao et al., 2020; Karras et al., 2020a; Cui et al., 2022; Jiang et al., 2021a; Chen et al., 2021) demonstrate that data insufficiency causes the overfitting of $D$, which deteriorates the training stability of GANs, compromising the quality and diversity of generated samples. To overcome this challenge, various approaches have been proposed, focusing on proposing novel augmentation techniques (Zhao et al., 2020; Karras et al., 2020a; Cui et al., 2022; Jiang et al., 2021a; Chen et al., 2021), designing better regularization for $D$ (Liu et al., 2021; Fang et al., 2022; Tseng et al., 2021), and applying pre-trained models (Sauer et al., 2021; Kumari et al., 2022) to $D$ to address the problem of discriminator overfitting. Nearly all of these methods aim to prevent $D$ from becoming overly confident in DE-GANs training, thereby maintaining the overlapping parts, and resulting in more useful gradients for $G$ during training.

### 2.3 Improve GANs Training via Critic Scores

In recent years, some studies (Azadi et al., 2018; Sinha et al., 2020) utilize the critic scores to improve the training of GANs. The widely-used one is Top-k GAN (Sinha et al., 2020). Top-k GAN proposes a linear decrease k (Linear Top-k) approach for the Top-k method in GANs and sets the initial value of k as the batchsize and the minimum value of k as **constant** $\times$ batchsize. In this case, the bad samples (low critic scores samples) can be thrown away during training, thus improving the training of GANs. The difference between the Linear Top-k and ATop-k is shown as follows. The value of k in Linear Top-k is gradually decreased with a fixed declining rate, and the minimum value of k is set as **constant** $\times$ batchsize. On the contrary, the value of k in the proposed ATop-k is controlled by the overfitting degree of $D$ adaptively (see Eq.(11)), with the possibility of being increased or decreased, and the minimum value of k is set as 1. ATop-k is more suited to data-efficient scenarios.

## 3 Methodology

### 3.1 Understanding the Training of DE-GANs

As pointed out in studies (Zhao et al., 2020; Karras et al., 2020a; Yang et al., 2022a), DE-GANs suffer from the overfitting of $D$ during training. In this section, we unveil a novel insight regarding the significance of the useful gradients of $G$ in DE-GANs training. Specifically, as the overfitting degree of $D$ increases, i.e., $D$ becomes increasingly confident in distinguishing real and fake data, the overlapping parts become gradually smaller (Karras et al., 2020a). The progressive disappearance of overlapping parts shows that the generated samples are further and further away from the real data distribution (Karras et al., 2020a), leading to the gradients of $G$ becoming progressively less useful (Sinha et al., 2020; Wu et al., 2019). At the point when $D$ becomes overly confident, i.e., $D$ reaches optimal as discussed in Arjovsky & Bottou (2017), the overlapping

parts disappear or can be ignored. Consequently, the disappearance of overlapping parts can cause the gradient instability or vanishing problem of $G$.

Without loss of generality, we select the widely-used DE-GAN backbones, i.e., StyleGAN2 (Karras et al., 2020b) and FastGAN (Liu et al., 2021), for analysis. Following the theory shown in Arjovsky & Bottou (2017) and Mescheder et al. (2018), for both StyleGAN2 with non-saturating loss and FastGAN with hinge loss (both with regularizations), there exists an optimal discriminator $D^*$ (Arjovsky & Bottou, 2017). Then, following this perfect discrimination theorems in (Arjovsky & Bottou, 2017), we can conclude that the optimal $D^*$ is overly confident in distinguishing both real and fake samples. In this case, the overlapping parts between $P_G$ and $P_R$ disappear or can be ignored.

Next, we theoretically analyze how the disappearance of overlapping parts can cause the gradient problem for the $G$ on both StyleGAN2 and FastGAN as follows:

**(1)** For StyleGAN2, the loss function is the non-saturating loss, formulated as

$$
\begin{aligned}
V_D(G, D) &= \mathbb{E}_{x \sim P_R}[\log D(x)] + \mathbb{E}_{x \sim P_G}[\log(1 - D(x))], \\
V_G(G, D) &= -\mathbb{E}_{x \sim P_G}[\log(D(x))].
\end{aligned}
\tag{1}
$$

Based on the theory of the original GAN (Goodfellow et al., 2014), we can obtain the optimal $D^*$ for Eq.(1) as

$$
D^*(x) = \frac{P_R(x)}{P_R(x) + P_G(x)}.
\tag{2}
$$

Following the perfect discrimination theorem, under this optimal $D^*$, the overlapping parts between $P_G$ and $P_R$ disappear or can be ignored. Then, based on Theorem 2.6 in Arjovsky & Bottou (2017), the complete disappearance of overlapping parts causes the gradient instability of $G$ during training.

**(2)** For FastGAN, the loss function is the hinge loss (Miyato et al., 2018; Zhang et al., 2019), formulated as

$$
\begin{aligned}
V_D(D, G) &= \mathbb{E}_{x \sim P_R}[\min(0, -1 + D(x))] + \mathbb{E}_{x \sim P_G}[\min(0, -1 - D(x))], \\
V_G(D, G) &= -\mathbb{E}_{x \sim P_G}[D(x)].
\end{aligned}
\tag{3}
$$

According to Miyato et al. (2018), Lim & Ye (2017), and Tran et al. (2017), optimizing the hinge loss is equivalent to minimizing the so-called reverse **KL** divergence item $\mathbf{KL}(P_G \| P_R)$. To prove that the hinge loss in FastGAN suffers from the gradient vanishing of $G$ when the overlapping parts completely disappear, we first follow Suh and Lim & Ye (2017) to analyze the optimal $D^*(x)$ with hinge loss as follows:

**Lemma 1.** For the hinge loss, the optimal discriminator $D^*(x)$ is given by

$$
D^*(x) = \text{sign}(P_R(x) - P_G(x)), \forall x \in M \cup P,
\tag{4}
$$

where sign() indicates the sign function that returns $+1$ for a non-negative input; $-1$, otherwise. $M$ and $P$ are the supports of $P_R$ and $P_G$, respectively. $D^*(x) = +1$ in the equality case $P_R(x) = P_G(x)$.

*Proof.* See Appendix A.1.

Under this optimal $D^*(x)$, we introduce Lemma 2 for $G$ as follows:

**Lemma 2.** Let $G_\theta(z)$ be a differentiable function that induces a distribution $P_G$. Let $D$ be a differentiable discriminator. For an optimal discriminator in Lemma 1 $D^* : \mathcal{X} \to (-\infty, +\infty)$, the perfect discrimination theorems in Arjovsky & Bottou (2017) is still satisfied. Let $\|D - D^*\| < \epsilon$ and $\mathbb{E}_{G_\theta(z) \sim P_G}[\|J_\theta G_\theta(z)\|_2^2] \leq M^2$ (Arjovsky & Bottou, 2017), then

$$
\left\| \bigtriangledown_\theta \mathbb{E}_{G_\theta(z) \sim P_G}[D(G_\theta(z))] \right\|_2 < M\epsilon.
\tag{5}
$$

*Proof.* See Appendix A.2.

Then, we have $\lim\limits_{\|D - D^*\| \longrightarrow 0} \nabla_\theta \, \mathbb{E}_{G_\theta(z) \sim P_G}[D(G_\theta(z))] = 0$, which indicates that the gradient of $G$ vanishes under $D^*(x)$.

**Discussion of Regularizations in StyleGAN2 and FastGAN.** Both StyleGAN2 and FastGAN apply regularization to $D$ to enhance the training of DE-GANs. To demonstrate the rigor and reasonableness of our insights in §3.1, we analyze that regularizations in both StyleGAN2 and FastGAN aim to avoid $D$ becoming overly confident, thereby maintaining the overlapping parts, resulting in more useful gradients for $G$. Therefore, regularizations do not influence the soundness of our insights under optimal $D^*(x)$.

**(1)** For StyleGAN2, the $R_1$ regularization (Mescheder et al., 2018) is applied to $D$. According to Mescheder et al. (2018), applying the $R_1$ regularization to StyleGAN2 can avoid $D$ exactly providing 1 for real data and 0 for fake data during training, thus preventing $D$ from reaching optimal $D^*$, effectively maintaining the overlapping parts, which results in more useful gradients for $G$.

**(2)** For FastGAN, reconstructive regularization is applied to $D$. Similar to the $R_1$ regularization in Style-GAN2, reconstructive regularization also aims to prevent $D$ from being overly confident, thus leading to more useful gradients for $G$.

Then, based on the theory in Mescheder et al. (2018), applying regularizations in GANs can still be convergent when the GANs are initialized sufficiently close to the equilibrium point. Consequently, there still exists an optimal $D^*$ for both StyleGAN2 and FastGAN in theory. As per Karras et al. (2020a), Zhao et al. (2020), and Liu et al. (2021), applying regularizations in DE-GANs can still suffer from overfitting under limited data settings, which indicates that such optimal $D^*$ also exists in practice.

## 3.2 Generalization Ability of the Proposed Novel Insight

Existing approaches apply various strategies for $D$ to improve the DE-GANs training. Among these studies, data augmentation (DA) is the widely-used approach for all these methods. To demonstrate the generalization ability of the proposed novel insight in §3.1, we connect it with DA and highlight that DA in DE-GANs can enlarge the overlapping parts in the initial training state by augmenting both real and fake data distributions, which avoids the $D$ becoming overly confident, thereby indirectly providing more useful gradients for $G$ during training. Notably, the selection of DA in DE-GANs should follow Karras et al. (2020a), Tran et al. (2021), and Zhao et al. (2020) to avoid the leaking issue (Karras et al., 2020a; Zhang et al., 2020; Zhao et al., 2021; Zhang et al., 2024). In this case, we first prove that for both StyleGAN2 and FastGAN, applying DA to both real and fake samples for $G$ and $D$ in DE-GANs results in the optimization between augmented fake data distribution ($P_G^T$) and augmented real data distribution ($P_R^T$), shown as follows:

**(1)** For StyleGAN2, the loss function of DA in DE-GANs is shown as follows

$$
\begin{aligned}
V_D(D, G) &= \mathbb{E}_{T(x) \sim P_R^T}[\log D(T(x))] + \mathbb{E}_{T(x) \sim P_G^T}[\log(1 - D(T(x)))], \\
V_G(D, G) &= -\mathbb{E}_{T(x) \sim P_G^T}[\log(D(T(x)))],
\end{aligned}
\tag{6}
$$

where $T$ is the applied DA in DE-GANs. Then, based on the theory of the original GAN (Goodfellow et al., 2014), we have that

$$
D^*(T(x)) = \frac{P_R^T(T(x))}{P_R^T(T(x)) + P_G^T(T(x))}.
\tag{7}
$$

Given the optimal $D^*(T(x))$, following Theorem 2.5 in Arjovsky & Bottou (2017), training the $G$ with augmented samples $T(x)$ under non-saturating loss can be expressed as

$$
V_G(D^*, G) = \mathbf{KL}(P_G^T \,\|\, P_R^T) - 2\mathbf{JS}(P_G^T \,\|\, P_R^T),
\tag{8}
$$

where **KL** and **JS** are the Kullback–Leibler and Jensen–Shannon divergence, respectively, Eq.(8) demonstrates that the DA applied to both real and fake samples for $G$ and $D$ in DE-GANs causes the optimization between $P_G^T$ and $P_R^T$.

**(2)** For FastGAN, the objective function with DA can be formulated as

$$V_D(D, G) = \mathbb{E}_{T(x) \sim P_R^T}[\min(0, -1 + D(T(x)))] + \mathbb{E}_{T(x) \sim P_G^T}[\min(0, -1 - D(T(x)))],$$
$$V_G(D, G) = -\mathbb{E}_{T(x) \sim P_G^T}[D(T(x))], \tag{9}$$

where $T$ is the applied DA. Then, based on Miyato et al. (2018), Lim & Ye (2017), and Tran et al. (2017), optimizing the hinge loss with augmented samples $T(x)$ is equivalent to minimizing the so-called reverse **KL** divergence item **KL**$(P_G^T \| P_R^T)$, indicating that DA applied to both real and fake samples for $D$ and $G$ causes the optimization between $P_G^T$ and $P_R^T$.

To assist with understanding how distributions $P_G^T$ and $P_R^T$ can enlarge the overlapping parts in the initial training state, we provide Lemma 3 as follows:

**Lemma 3.** For two supports of the distributions $P_G$ and $P_R$ consisting of overlapping parts, if there exists one sample $x \in P_G \cap P_R$, for $\forall$ augmentation $T$, $T(x) \in P_G^T \cap P_R^T$.

*Proof.* See Appendix A.3

Lemma 3 shows that with the $T$ in DE-GANs, if the sample $x$ is in the overlapping parts between the supports of $P_G$ and $P_R$, $T(x)$ can still be in the overlapping parts between the supports of $P_G^T$ and $P_R^T$. Then, according to Tran et al. (2021), Zhao et al. (2020), and Karras et al. (2020a), DA in DE-GANs is applied with random parameters. Therefore, applying DA to one sample $x$ can produce a series of different augmented samples $T(x)$. As a result, DA in DE-GANs can yield overlapping parts between the supports of $P_G^T$ and $P_R^T$ consisting of more different samples $T(x)$. Consequently, the supports of $P_G^T$ and $P_R^T$ consist of more overlapping parts than the supports of $P_G$ and $P_R$ in the initial training state, which results in more generated samples closer to the real data distribution, thereby providing more useful gradients for $G$ during training, thus benefiting the training of DE-GANs.

### 3.3 Proposed Method

Driven by the insight in §3.1, different from the existing approaches that enhance the $D$ in DE-GANs training to obtain more useful gradients for $G$ during training, we address the DE-GANs training task from a novel perspective, i.e., directly providing more useful gradients for $G$ through selecting fake samples to update $G$ during training. This selection process is guided by the critic scores (Azadi et al., 2018; Sinha et al., 2020), i.e., the discriminator outputs, on the fake samples. To this end, we propose a general training strategy called adaptive Top-k (ATop-k) for DE-GANs, as shown in Figure 2. Specifically, only the Top-k high-score fake samples, i.e., realistic-looking fake samples closer to the real data distribution, are used to update the generator ($G$). The more overfitting of $D$, the less overlapping parts can be observed during training, resulting in fewer generated samples having more useful gradients. Therefore, the value of k in ATop-k is adjusted adaptively based on the degree of overfitting without manually tuning. Specifically, following ANDA (Zhang et al., 2024), APA (Jiang et al., 2021a) and ADA (Karras et al., 2020a), we apply an overfitting heuristic $\eta$ that quantifies the degree of $D$'s overfitting as follows:

$$\eta = \mathbb{E}(\text{sign}(D_{real})), D_{real} = \text{logit}(D(x)), \tag{10}$$

where sign() means the sign function that returns $+1$ for the non-negative input; $-1$, otherwise. We follow the ADA, APA and ANDA to use $\eta$ to obtain the overfitting probability $p$. Specifically, we initialize $p$ to zero and adjust its value once every four minibatches[1] based on the $\eta$. If $\eta$ is larger/smaller than a pre-set threshold $t$ (in most cases of our experiments, $t = 0.6$), i.e., $\eta$ indicates too much/little overfitting regarding $t$, the probability $p$ will be increased/decreased by one fixed step[2]. Then, $p$ can increase from zero to one in 500k images shown to $D$. After every adjustment, we clamp $p$ from below to 0. The larger value of $p$ means that the more overfitting degree. Then, we set the initial k value as batchsize, and adjust its value based on the $p$ as follows:

---

[1]This choice follows from the official codes of StyleGAN2 + ADA
[2]The step size follows from the StyleGAN2 + ADA

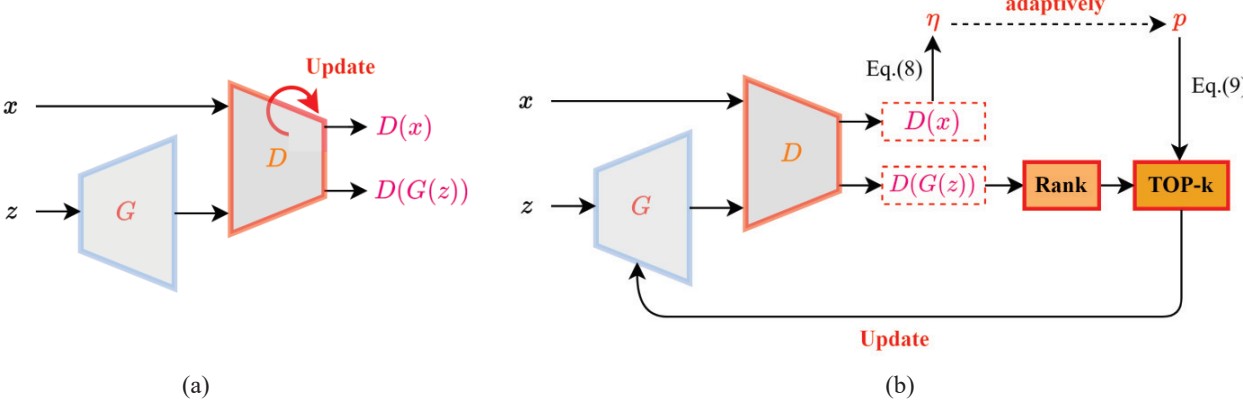

(a)                                                           (b)

Figure 2: An overview of adaptive Top-k (ATop-k) for (a) updating $D$ and (b) updating $G$ in DE-GANs. For updating $D$, both real and fake samples are used to update the parameters of $D$. For updating $G$, only the Top-k high-score fake samples are applied to update the parameters of $G$. Specifically, the value of $k$ is controlled by the overfitting probability $p$ adaptively by Eq.(11). *Best viewed in color.*

$$k = \begin{cases} k-1 & (p) \\ k+1 & (1-p) \end{cases} \tag{11}$$

where k is adjusted to k-1 under the probability $p$ and is adjusted to k+1 under the probability $1-p$. We set the range of k as $k_{min} = 1$ and $k_{max} =$ batchsize. Then, the ATop-k in the DE-GANs can be formulated as:

$$\begin{aligned} V_D(G, D) &= \mathbb{E}_{x \sim P_R}[\tilde{f}_1(D(x))] + \mathbb{E}_{x \sim P_G}[\tilde{f}_2(D(x))], \\ V_G(G, D) &= -\mathbb{E}_{x \sim P_G}[\tilde{g}(\text{Top-k}(D(x)))], \end{aligned} \tag{12}$$

where $\tilde{f}_1$, $\tilde{f}_2$ and $\tilde{g}$ are scalar-to-scalar functions. For the StyleGAN2 backbone, $\tilde{f}_1(x) = \log(x)$, $\tilde{f}_2(x) = \log(1-x)$ and $\tilde{g}(x) = \log(x)$. For the FastGAN backbone, $\tilde{f}_1(x) = \min(0, -1+x)$, $\tilde{f}_2(x) = \min(0, -1-x)$ and $\tilde{g}(x) = x$.

We conduct several experiments to better demonstrate that ATop-k can enable $G$ to directly obtain more useful gradients during training, thus leading to better performance. Because the distributions observed by $D$ can effectively show how close the fake data distribution is to the real data distribution, we observe the output of $D$ in the experiments to show how useful the gradients of $G$ are during training. Based on Jiang et al. (2021a), Karras et al. (2020a) and Cui et al. (2022), the less difference between the output of $D$ on real images $x$ and on fake images $G(z)$, i.e., smaller $D(x) - D(G(z))$, indicates that the generated distribution is closer to the real data distribution, resulting in more useful gradients for $G$ during training. Therefore, we compare StyleGAN2 + ATop-k with StyleGAN2, StyleGAN2 + Linear Top-k (Sinha et al., 2020) and StyleGAN2 + Worst-k on the 100-shot Obama dataset by $D(x) - D(G(z))$, where Linear Top-k is introduced in Sinha et al. (2020) and Worst-k is a training strategy that selects the k lowest-score fake samples to update the $G$. As shown in Figure 3 (a), adding Worst-k to StyleGAN2 increases the value of $D(x) - D(G(z))$, demonstrating that Worst-k results in less useful gradients for $G$ during training. In contrast, adding ATop-k to StyleGAN2 can clearly decrease the value of $D(x) - D(G(z))$, indicating that ATop-k can enable $G$ to obtain more useful gradients during training. Moreover, ATop-k also achieves a significant improvement compared to Linear Top-k in StyleGAN2, showing that the ATop-k is more suitable for DE-GANs. The FID results comparing ATop-k and Worst-k are shown in Figure 3 (b). It is evident that more useful gradients for $G$ lead to better performance. More comparison results between ATop-k and Linear Top-k can be found in the ablation study (refer to §4.4).

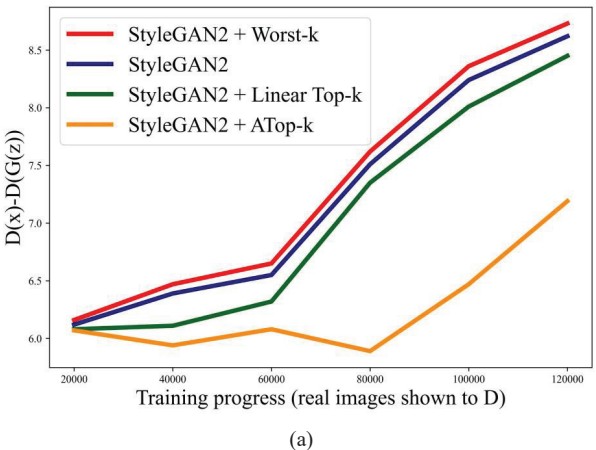

| Method | FID |
|---|---|
| StyleGAN2 | 65.57 |
| StyleGAN2 + Worst-k (k = 3) | 68.71 |
| StyleGAN2 **+ ATop-k** | **55.78** |
| StyleGAN2 + ADA | 45.69 |
| StyleGAN2 + ADA + Worst-k (k = 3) | 53.67 |
| StyleGAN2 + ADA **+ ATop-k** | **40.79** |

(a)

(b)

Figure 3: Illustration Experiments of ATop-k can provide more useful gradients for $G$ during training. (a) The comparison of the value of $D(x) - D(G(z))$ for StyleGAN2, StyleGAN2 + Linear Top-k, StyleGAN2 + Worst-k and StyleGAN2 + ATop-k on the 100-shot Obama dataset during training. According to Jiang et al. (2021a) and Karras et al. (2020a), the smaller $D(x) - D(G(z))$ indicates that the generated distribution is closer to the real data distribution, resulting in more useful gradients for $G$ during training. Adding Worst-k (using k lowest-score fake samples to update $G$) to StyleGAN2 can increase the value of $D(x) - D(G(z))$, demonstrating that Worst-k causes less useful gradients for $G$ during training. In contrast, adding ATop-k to StyleGAN2 can clearly decrease the value of $D(x) - D(G(z))$, indicating that ATop-k can enable more useful gradients for $G$ during training. Furthermore, ATop-k also achieves a clear improvement compared with Linear Top-k (Sinha et al., 2020), showing that the design of ATop-k is more suitable for DE-GANs. (b) Fréchet Inception Distance (FID) (Heusel et al., 2017) score on 100-shot Obama dataset. The less useful gradients for $G$ during training caused by Worst-k can finally decrease performance. On the contrary, the more useful gradients for $G$ caused by ATop-k can achieve a significant improvement.

## 4 Experiment

We show the superiority of ATop-k by adding it to five different DE-GANs and evaluating it on commonly-used datasets. More details of experiments and the anonymous link of the pre-trained model with test code can be found in Appendix B.1.

### 4.1 Datasets and Implementation Details

We select low-shot (Zhao et al., 2020) and FFHQ (Karras et al., 2020b) datasets for the experiments. According to Zhao et al. (2020), all the images are resized to $256 \times 256$ for a fair comparison. Low-shot datasets contain five datasets (i.e., Obama, Grumpy Cat, Panda, Animal-Face Cat, and Animal-Face Dog) with 100, 100, 100, 160, and 389 training images, respectively. The FFHQ dataset contains 70K high-resolution images of human faces. Following FakeCLR (Li et al., 2022a), we select a subset of 100, 1K, 2K, and 5K for a fair comparison. We demonstrate the superiority of the proposed ATop-k using five different DE-GANs, i.e., StyleGAN2, StyleGAN2 + Diff-Augment, StyleGAN2 + ADA, InsGen + ADA and Diffusion-Projected GAN. Specifically, the StyleGAN2 method does not apply any DA methods during training, and Diffusion-Projected GAN applies the FastGAN as the backbone. The commonly used Fréchet Inception Distance (FID) (Heusel et al., 2017), and Precision and Recall (Kynkäänniemi et al., 2019) are applied as the evaluation metrics. Following prior work (Zhao et al., 2020; Karras et al., 2020a), the full dataset is used as the reference distribution for all metrics calculation. The best FID result for each method is reported in the experiments.

| Method | MA | P | 100-shot | | | Animal-Face | |
|---|---|---|---|---|---|---|---|
| | | | Obama | Grumpy Cat | Panda | Cat | Dog |
| Scale/shift (Noguchi & Harada, 2019) | No | Yes | 50.72 | 34.20 | 21.38 | 54.83 | 83.04 |
| MineGAN (Wang et al., 2020) | No | Yes | 50.63 | 34.54 | 14.84 | 54.45 | 93.03 |
| TransferGAN (Wang et al., 2018) | No | Yes | 48.73 | 34.06 | 23.20 | 52.61 | 82.38 |
| TransferGAN + DA (Zhao et al., 2020) | Yes | Yes | 39.85 | 29.77 | 17.12 | 49.10 | 65.57 |
| FreezeD (Mo et al., 2020) | No | Yes | 41.87 | 31.22 | 17.95 | 47.70 | 70.46 |
| StyleGAN2 (Karras et al., 2020b) | No | No | 80.20 | 48.90 | 34.27 | 71.71 | 131.90 |
| FakeCLR (Li et al., 2022a) | Yes | No | 26.95 | 19.56 | 8.42 | 26.34 | 42.02 |
| FastGAN (Liu et al., 2021) | Yes | No | 35.80 | 25.75 | 9.70 | 33.85 | 52.46 |
| FreGAN (Yang et al., 2022b) | Yes | No | 33.39 | 24.93 | 8.97 | 31.05 | 47.85 |
| AugSelf-StyleGAN2 (Hou et al., 2023) | Yes | No | 26.00 | 19.81 | 8.36 | 30.53 | 48.19 |
| Projected GAN (Sauer et al., 2021) | Yes | Yes | 11.21 | 15.80 | 3.98 | 18.01 | 17.88 |
| EDM (Karras et al., 2022) | Yes | No | 51.30 | 36.90 | 23.70 | 48.60 | 100.10 |
| EDM + DA (Karras et al., 2022) | Yes | No | 37.10 | 29.94 | 10.81 | 36.88 | 57.14 |
| Patch Diffusion (Wang et al., 2023a) | Yes | No | 41.47 | 30.89 | 13.25 | 43.71 | 72.17 |
| StyleGAN2* (Karras et al., 2020b) | Yes | No | 65.57 | 39.92 | 25.08 | 51.66 | 87.96 |
| **+ ATop-k** | Yes | No | **55.78** | **33.22** | **21.79** | **45.44** | **77.47** |
| StyleGAN2 + Diff-Augment (Zhao et al., 2020) | Yes | No | 46.87 | 27.08 | 12.06 | 42.44 | 58.85 |
| **+ ATop-k** | Yes | No | **40.96** | **26.22** | **10.28** | **37.74** | **51.29** |
| StyleGAN2 + ADA (Karras et al., 2020a) | Yes | No | 45.69 | 26.62 | 12.90 | 40.77 | 56.83 |
| **+ ATop-k** | Yes | No | **40.79** | **26.31** | **11.93** | **37.91** | **53.35** |
| InsGen + ADA (Yang et al., 2021) | Yes | No | 32.42 | 22.01 | 9.85 | 33.01 | 44.93 |
| **+ ATop-k** | Yes | No | **25.27** | **19.32** | **7.92** | **23.02** | **35.37** |
| Diffusion-Projected GAN (Wang et al., 2023b) | Yes | Yes | 10.54 | 15.13 | 3.39 | 17.86 | 17.22 |
| **+ ATop-k** | Yes | Yes | **9.92** | **14.36** | **3.27** | **17.01** | **16.74** |

Table 1: FID (Heusel et al., 2017) scores (lower is better) of different methods on $256 \times 256$ low-shot datasets. MA means Massive Augmentation, i.e., xflipping, which consists of the same meaning in Genco (Cui et al., 2022). For a fair comparison, FID is measured using 5K generated samples with the whole training dataset as the reference distribution. The FIDs are averaged over three runs; all standard deviations are less than 1% relatively. The results of StyleGAN2* with MA, Projected GAN and Diffusion-Projected GAN are obtained by running the official open source codes. The results of EDM are directly from Aghabozorgi et al. (2023), and the results of EDM + DA and Patch Diffusion are obtained by running the official open-source codes. The results of the different GAN baseline methods are as reported in FakeCLR (Li et al., 2022a) and FreGAN (Yang et al., 2022b) for the StyleGAN2 and FastGAN backbones, respectively. Here, **P** represents **Pre-training**.

| Method | Obama | | Grumpy Cat | | Panda | | AF-Cat | | AF-Dog | |
|---|---|---|---|---|---|---|---|---|---|---|
| | **P** | **R** | **P** | **R** | **P** | **R** | **P** | **R** | **P** | **R** |
| Diffusion-Projected GAN | 0.999 | 0.240 | 0.996 | 0.149 | 0.999 | 0.410 | 0.997 | 0.182 | 0.998 | 0.203 |
| **+ATop-k** | **0.999** | **0.247** | **0.997** | **0.152** | **0.999** | **0.422** | **0.998** | **0.185** | **0.998** | **0.208** |

Table 2: The comparison of Precision and Recall (higher is better) with Diffusion-Projected GAN (Wang et al., 2023b) and Diffusion-Projected GAN + ATop-k on the $256 \times 256$ low-shot datasets. The Precision (**P**) and Recall (**R**) are averaged over five runs; all standard deviations are less than 1% relatively.

According to Parmar et al. (2022), the data-resized method can influence the performance of GANs. For a fair comparison, we follow the official open-source codes[3] for preprocessing and resizing the FFHQ dataset. For

---

[3] https://github.com/NVlabs/stylegan2-ada-pytorch

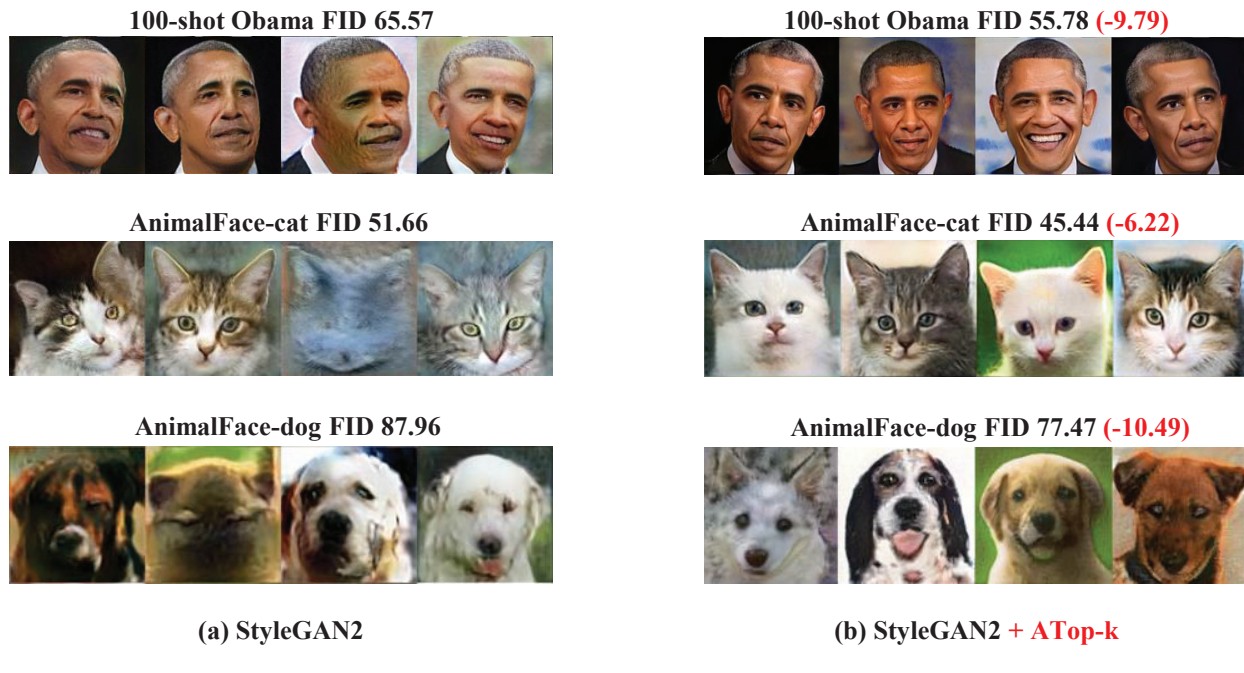

**100-shot Obama FID 65.57**

**100-shot Obama FID 55.78 (-9.79)**

**AnimalFace-cat FID 51.66**

**AnimalFace-cat FID 45.44 (-6.22)**

**AnimalFace-dog FID 87.96**

**AnimalFace-dog FID 77.47 (-10.49)**

**(a) StyleGAN2**

**(b) StyleGAN2 + ATop-k**

Figure 4: The comparison of generated results with StyleGAN2 and StyleGAN2 + ATop-k on 100-shot Obama, AnimalFace-cat and AnimalFace-dog datasets. (a) Images generated by StyleGAN2. (b) Images generated by StyleGAN2 + ATop-k. The decreasing value of FID in red color demonstrates the improvement of StyleGAN2 + ATop-k compared with baseline StyleGAN2. *Best viewed in color.*

the experiments on low-shot datasets with StyleGAN2, StyleGAN + ADA and StyleGAN2 + Diff-Augment, we set the batchsize as 4 and the overall training duration as 2000k images. For the experiments on low-shot datasets with InsGen, we set the batchsize as 64 and the overall training duration as 25000k images. For the experiments on low-shot datasets with Diffusion-Projected GAN, we set the batchsize as 64 and the overall training duration as 100000k images. For the experiments on FFHQ datasets, we set the batchsize as 64 and the overall training duration as 25000k images for all methods.

## 4.2 Results on Low-shot Datasets

The results on $256 \times 256$ low-shot datasets are shown in Table 1. The proposed ATop-k improves the performance of all DE-GANs, which demonstrates the generalization ability of ATop-k. More importantly, Diffusion-Projected GAN applied DA, regularizations and pre-trained models to achieve state-of-the-art performance on the low-shot datasets; adding ATop-k can further improve the performance. In addition, following Aghabozorgi et al. (2023), we also report the results of several state-of-the-art diffusion models in Table 1. It is clear that DE-GANs outperform diffusion models on low-shot datasets. To further show the effectiveness of ATop-k, the compared Precision and Recall of adding ATop-k to Diffusion-Projected GAN are shown in Table 2. ATop-k can further increase the Precision and Recall for Diffusion-Projected GAN. Additionally, the compared generated images on StyleGAN2 and StyleGAN2 + ATop-k are shown in Figure 4. More generated images on low-shot datasets can be found in Appendix B.2.

## 4.3 Results on FFHQ Dataset

The results on the FFHQ dataset ($256 \times 256$) are shown in Table 3. Following FakeCLR (Li et al., 2022a), we conduct experiments on the FFHQ 100, FFHQ 1K, FFHQ 2K and FFHQ 5K subsets. The proposed ATop-k increases the performance using five different DE-GANs, indicating the generalization ability of the proposed ATop-k. Additionally, the comparison of Precision and Recall with Diffusion-Projected GAN and

| Method | MA | Pre-training | FFHQ | | | |
|---|---|---|---|---|---|---|
| | | | 100 | 1K | 2K | 5K |
| APA (Jiang et al., 2021a) | Yes | No | 65.00 | 18.89 | 16.90 | 8.38 |
| AugSelf-StyleGAN2+ (Hou et al., 2023) | Yes | No | - | 20.39 | - | 9.15 |
| FakeCLR (Li et al., 2022a) | Yes | No | 42.56 | 15.92 | 9.90 | 7.25 |
| Projected GAN (Sauer et al., 2021) | Yes | Yes | 26.25 | 11.12 | 8.25 | 6.85 |
| EDM (Karras et al., 2022) | Yes | No | 79.10 | - | - | - |
| EDM + DA (Karras et al., 2022) | Yes | No | 50.73 | 30.75 | 27.17 | 24.51 |
| Patch Diffusion (Wang et al., 2023a) | Yes | No | 44.45 | 28.03 | 25.32 | 22.63 |
| StyleGAN2 (Karras et al., 2020b) | No | No | 179.00 | 100.16 | 54.30 | 49.68 |
| **+ATop-k** | No | No | **121.96** | **66.37** | **40.98** | **21.75** |
| StyleGAN2 + Diff-Augment (Zhao et al., 2020) | Yes | No | 61.91 | 25.66 | 24.32 | 10.45 |
| **+ATop-k** | Yes | No | **55.24** | **22.98** | **16.12** | **10.15** |
| StyleGAN2 + ADA (Karras et al., 2020a) | Yes | No | 85.8 | 21.29 | 15.39 | 10.96 |
| **+ATop-k** | Yes | No | **69.92** | **20.71** | **15.09** | **10.71** |
| InsGen (Yang et al., 2021) | Yes | No | 45.75 | 18.21 | 11.47 | 7.83 |
| **+ATop-k** | Yes | No | **43.09** | **16.97** | **10.52** | **7.40** |
| Diffusion-Projected GAN (Wang et al., 2023b) | Yes | Yes | 25.47 | 8.76 | 7.99 | 6.59 |
| **+ATop-k** | Yes | Yes | **24.32** | **8.24** | **7.89** | **6.51** |

Table 3: FID scores (lower is better) of different methods on the $256 \times 256$ FFHQ dataset. Following FakeCLR (Li et al., 2022a), the experiments are performed on 100, 1K, 2K and 5K subsets of the FFHQ dataset. Massive Augmentation (MA) (Cui et al., 2022), i.e., xflipping, is applied in all of the methods. For a fair comparison, the FIDs are averaged over five runs; all standard deviations are less than 1% relatively. The results of the Projected GAN and Diffusion-Projected GAN are obtained by ourselves by running the official open-source codes. The results of EDM are directly from Aghabozorgi et al. (2023), and the results of EDM + DA and Patch Diffusion are obtained by ourselves via running the official open-source codes.

| Method | FFHQ-100 | | FFHQ-1K | | FFHQ-2K | | FFHQ-5K | |
|---|---|---|---|---|---|---|---|---|
| | **P** | **R** | **P** | **R** | **P** | **R** | **P** | **R** |
| Diffusion-Projected GAN | 0.737 | 0.002 | 0.720 | 0.189 | 0.713 | 0.243 | 0.710 | 0.299 |
| **+ATop-k** | **0.746** | **0.003** | **0.725** | **0.206** | **0.716** | **0.244** | **0.710** | **0.301** |

Table 4: The comparison of Precision and Recall (higher is better) with Diffusion-Projected GAN (Wang et al., 2023b) and Diffusion-Projected GAN + ATop-k on the $256 \times 256$ FFHQ dataset. The Precision (**P**) and Recall (**R**) are averaged over five runs; all standard deviations are less than 1% relatively.

Diffusion-Projected GAN + ATop-k are shown in Table 4. Adding ATop-k can improve the performance compared with Diffusion-Projected GAN, which further demonstrates the effectiveness of the proposed ATop-k. To further show the superiority of ATop-k, the images generated on the FFHQ dataset can be found in Appendix B.2.

## 4.4 Comparative Experiments

**ATop-k v.s. Fixed Top-k.** To show the effectiveness of the proposed ATop-k, we conduct an ablation study by selecting different fixed Top-k values on the 100-shot Obama dataset with StyleGAN2 + Diff-Augment method. We select fixed values of k as 3, 5, and 10, respectively, and the results are shown in Table 5. Although applying fixed Top-k in DE-GANs can throw away the bad samples during training, selecting lower values of k in the early training stage can lead to $G$ being updated with fewer meaningful fake samples, thus leading to undesired performance.

| StyleGAN2 + Diff-Augment | FID (100-shot Obama) |
|---|---|
| + fixed Top-k (k=3) | 47.14 |
| + fixed Top-k (k=5) | 45.91 |
| + fixed Top-k (k=10) | 45.78 |
| **+ ATop-k** | **40.96** |

Table 5: Experiments of applying different fixed Top-k in DE-GANs. We select fixed values of k as 3, 5 and 10 on the 100-shot Obama dataset with StyleGAN2 + Diff-Augment for the ablation study. The FIDs are averaged over three runs, all standard deviations are less than 1% relatively.

| Method | FID |
|---|---|
| StyleGAN2 + Linear Top-k | 63.41 |
| StyleGAN2 **+ ATop-k** | **55.78** |
| StyleGAN2 + Diff-Augment + Linear Top-k | 44.87 |
| StyleGAN2 + Diff-Augment **+ ATop-k** | **40.96** |
| Diffusion-Projected GAN + Linear Top-k | 10.46 |
| Diffusion-Projected GAN **+ ATop-k** | **9.92** |

Table 6: FID scores (lower is better) of different methods on the $256 \times 256$ 100-shot Obama dataset. We select StyleGAN-based and FastGAN-based methods for comparison. The FIDs are averaged over three runs, all standard deviations are less than 1% relatively.

**ATop-k v.s. Linear Top-k.** Top-k GAN (Sinha et al., 2020) proposes a linear decrease k approach for the Top-k method in GANs and sets the minimum value of k as the $0.75 \times$ batchsize. We also compare the proposed ATop-k with this Linear Top-k strategy on the 100-shot Obama dataset with different DE-GANs methods, and the results are shown in Table 6. According to Figure 3 and Table 6, although the Linear Top-k strategy can slightly benefit the training of DE-GANs, it cannot effectively avoid $D$ being overly confident. In contrast, ATop-k can better prevent $D$ from becoming overly confident (see Figure 3 (a)), thus maintaining the overlapping parts and leading to better performance.

## 5 Conclusion

In this paper, we unveil a novel insight in DE-GANs training, regarding the significance of useful gradients of $G$. To further understand this novel insight, we apply this to existing methods and point out that these methods focus on $D$ to prevent $D$ from becoming overfitting thus indirectly resulting in more useful gradients for $G$ during training. Driven by this novel insight, we propose a simple yet effective general training strategy for DE-GANs to directly provide more useful gradients for $G$ by selecting fake samples to update $G$ during training, namely adaptive Top-k (ATop-k). Extensive experiments on several datasets demonstrate that ATop-k can effectively improve the training of DE-GANs and achieve better performance on different DE-GANs backbones.

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

## A  Proofs

### A.1  The Proof of Lemma 1

**Lemma 1.** For the hinge loss in FastGAN, the optimal discriminator $D^*(x)$ is given by

$$D^*(x) = \text{sign}(P_R(x) - P_G(x)), \forall x \in M \cup P, \tag{13}$$

where sign() indicates the sign function that returns $+1$ for a non-negative input; $-1$, otherwise. $M$ and $P$ are the supports of $P_R$ and $P_G$, respectively. $D^*(x) = +1$ in the equality case $P_R(x) = P_G(x)$.

**Proof.** For hinge loss, the discriminator aims to maximize

$$
\begin{aligned}
V_D(D, G) = &\mathbb{E}_{x \sim P_R}[\min(0, -1 + D(x))] \\
&+ \mathbb{E}_{x \sim P_G}[\min(0, -1 - D(x))].
\end{aligned}
\tag{14}
$$

Let $f(D) = a[\min(0, -1 + D)] + b[\min(0, -1 - D)]$, where $D = D(x)$ is the output of discriminator, $a = P_R(x)$ and $b = P_G(x)$. According to Geometric GAN (Lim & Ye, 2017), hinge loss in GANs replaces the label $y$ in the hinge loss of SVM as $D$. In hinge loss, label $y > 1$ and $y < -1$ can not provide the gradient for the loss function. Thus, considering with non-negative loss value in theory, the value of $D$ in hinge loss is $-1 \leq D \leq 1$. Then, there exist the following cases.

**Case 1:** $a > b$. In this case, $f(D) = -a - b + (a - b)D$, because $a > b$, then $\underset{D}{argmax} f(D) = 1$ for maximizing the $f(D)$.

**Case 2:** $a = b$. Based on Theorem 3.1 in the Geometric GAN (Lim & Ye, 2017), then $\underset{D}{argmax} f(D) = 1$ for maximizing the $f(D)$.

**Case 3:** $a < b$. In this case, $f(D) = -a - b + (a - b)D$, because $a < b$, then $\underset{D}{argmax} f(D) = -1$ for maximizing the $f(D)$.

To conclude, $\underset{D}{argmax} f(D) = \text{sign}(a - b)$. Notice that for each $x$, $P_R(x)$ and $P_G(x)$ correspond to $a$ and $b$, respectively. Then, we can obtain that

$$D^*(x) = \text{sign}(P_R(x) - P_G(x)), \forall x \in M \cup P, \tag{15}$$

where the equality case $P_R(x) = P_G(x)$ yields $D^*(x) = +1$. That concludes our proof.  □

### A.2  The Proof of Lemma 2

**Lemma 2.** Let $G_\theta(z)$ be a differentiable function that induces a distribution $P_G$. Let $D$ be a differentiable discriminator. For an optimal discriminator in Lemma 1 $D^* : \mathcal{X} \to (-\infty, +\infty)$, the perfect discrimination theorems in Arjovsky & Bottou (2017) is still satisfied. Let $\|D - D^*\| < \epsilon$ and $\mathbb{E}_{G_\theta(z) \sim P_G}[\|J_\theta G_\theta(z)\|_2^2] \leq M^2$ (Arjovsky & Bottou, 2017), then

$$\left\| \nabla_\theta \mathbb{E}_{G_\theta(z) \sim P_G}[D(G_\theta(z))] \right\|_2 < M\epsilon. \tag{16}$$

**Proof.** With the optimal discriminator $D^*$ in Lemma 1, for $\forall x \in (-\infty, +\infty)$, if the $P_G$ and $P_R$ consist of the support contained on two disjoint compact subsets, $D^*(x)$ has accuracy 1 and $\nabla_x D^*(x) = 0$ (this is because the $\nabla_x \text{sign}(x) \equiv 0$). In this case, the perfect discrimination theorems in Arjovsky & Bottou (2017) is still satisfied under $D^* : \mathcal{X} \to (-\infty, +\infty)$. Then, following the $\|D\|$ denoted in §2.2.1 as in Arjovsky & Bottou (2017), then, using Jensen's inequality and the chain rule on this support, we have

$$\left\|\bigtriangledown_\theta \mathbb{E}_{G_\theta(z) \sim P_G}[D(G_\theta(z))]\right\|_2^2$$
$$\leq \mathbb{E}_{G_\theta(z) \sim P_G}[\|\bigtriangledown_\theta D(G_\theta(z))\|_2^2]$$
$$\leq \mathbb{E}_{G_\theta(z) \sim P_G}[\|\bigtriangledown_x D(G_\theta(z))\|_2^2 \|J_\theta G_\theta(z)\|_2^2]$$
$$< \mathbb{E}_{G_\theta(z) \sim P_G}[(\|\bigtriangledown_x D^*(G_\theta(z))\|_2 + \epsilon)^2 \|J_\theta G_\theta(z)\|_2^2] \tag{17}$$
$$= \mathbb{E}_{G_\theta(z) \sim P_G}[\|J_\theta G_\theta(z)\|_2^2 \epsilon^2]$$
$$\leq M^2 \epsilon^2,$$

where $\|J_\theta G_\theta(z)\|_2$ is the norm of the Jacobian matrix which depends on the $P_G$ and is defined as in Arjovsky & Bottou (2017).

Taking square root of Eq.(17), we then get

$$\left\|\bigtriangledown_\theta \mathbb{E}_{G_\theta(z) \sim P_G}[D(G_\theta(z))]\right\|_2 < M\epsilon. \tag{18}$$

That concludes our proof. $\square$

### A.3 The Proof of Lemma 3

**Lemma 3.** For two supports of the distributions $P_G$ and $P_R$ consisting of overlapping parts, if there exists one sample $x \in P_G \cap P_R$, for $\forall$ augmentation $T$, $T(x) \in P_G^T \cap P_R^T$.

***Proof.*** Based on §3.2 in the main paper, applying data augmentation in DE-GANs transforms the distribution of generated samples ($P_G$) and the distribution of real samples ($P_R$) to augmented distributions $P_G^T$ and $P_R^T$ with a certain augmentation $T$. Thus, we can conclude that $[x \in P_G \cap P_R] \Rightarrow [x \in P_G$ and $x \in P_R] \Rightarrow [T(x) \in P_G^T$ and $T(x) \in P_R^T] \Rightarrow [T(x) \in P_G^T \cap P_R^T]$. That concludes our proof. $\square$

## B Experiment

### B.1 More details on the Experiments

**Experimental setup.** (1) The StyleGAN2-based results in the main paper are trained by an NVIDIA DGX workstation with four Tesla V100 ($4 \times 32G$) GPUs. The operating system is CentOS 7. To replicate our experimental environment, we recommend referring to the official open-source codes[3] for instructions in order to obtain the necessary software and Python libraries. (2) The FastGAN-based results in the main paper are trained by a workstation with CPU i9-10980XE, 128G ECC memory and four TITAN RTX GPUs ($4 \times 24G$). The operating system is Ubuntu 18.04. To replicate our experimental environment, we recommend referring to the official open-source codes[4] for instructions in order to obtain the necessary software and Python libraries.

**Pretrained models with test codes.** The pre-trained Diffusion-Projected GAN + ATop-k model with test code can be found in the anonymous Google Drive link[5]. We will release all training codes once the paper is accepted.

### B.2 More Generated Images

According to the main paper, more generated results on low-shot datasets with StyleGAN2 + Diff-Augment + ATop-k, StyleGAN2 + ADA + ATop-k, InsGen + ATop-k and Diffusion-Projected GAN + ATop-k are shown in Figures 5, 6, 7 and 8, respectively. More generated results on FFHQ datasets with StyleGAN2 + ATop-k, StyleGAN2 + Diff-Augment + ATop-k, StyleGAN2 + ADA + ATop-k, InsGen + ATop-k and Diffusion-Projected GAN + ATop-k are shown in Figures 9, 10, 11, 12 and 13, respectively.

---

[4]https://github.com/autonomousvision/projected-gan
[5]https://drive.google.com/file/d/176adFIzD3iq9-FQE1_F1WDL4xk12PKn-/view?usp=sharing

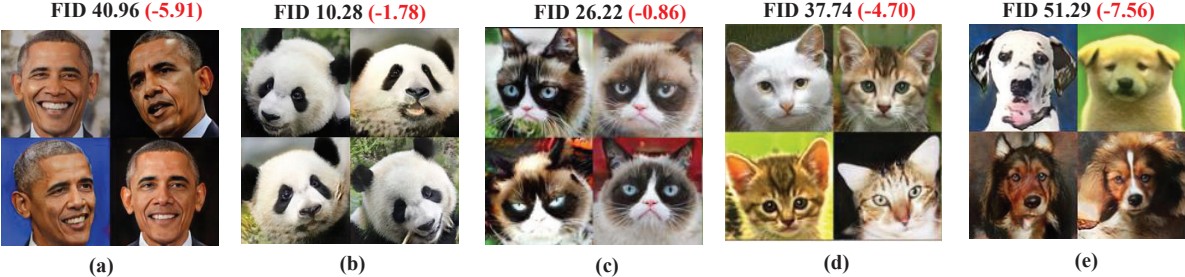

Figure 5: Images generated by StyleGAN2 + Diff-Augment + ATop-k on (a) 100-shot Obama dataset, (b) 100-shot Panda dataset, (c) 100-shot Grumpy-cat dataset, (d) AnimalFace-cat dataset and (e) AnimalFace-dog dataset. The decreasing value of FID in red color demonstrates the improvement of StyleGAN2 + Diff-Augment + ATop-k compared with baseline StyleGAN2 + Diff-Augment. *Best viewed in color.*

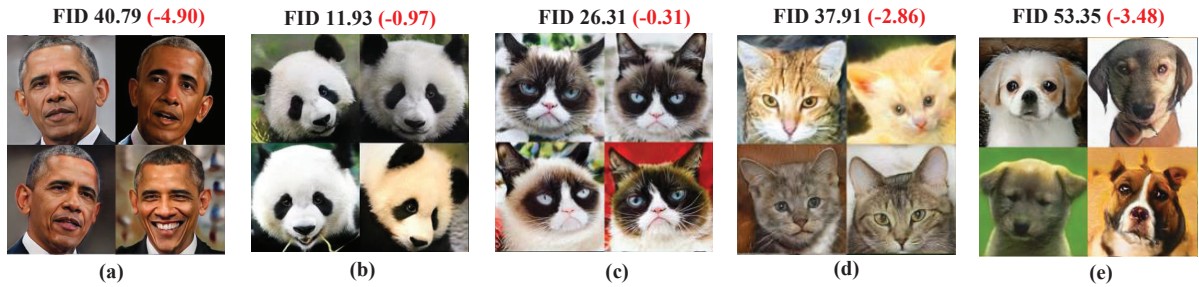

Figure 6: Images generated by StyleGAN2 + ADA + ATop-k on (a) 100-shot Obama dataset, (b) 100-shot Panda dataset, (c) 100-shot Grumpy-cat dataset, (d) AnimalFace-cat dataset and (e) AnimalFace-dog dataset. The decreasing value of FID in red color demonstrates the improvement of StyleGAN2 + ADA + ATop-k compared with baseline StyleGAN2 + ADA. *Best viewed in color.*

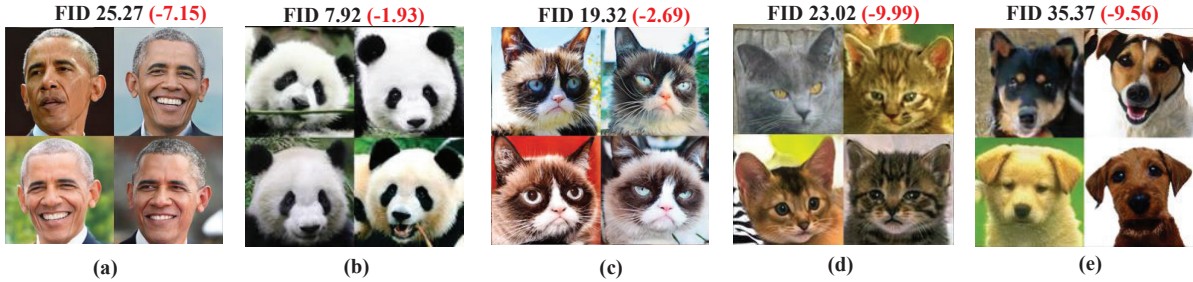

Figure 7: Images generated by InsGen + ATop-k on (a) 100-shot Obama dataset, (b) 100-shot Panda dataset, (c) 100-shot Grumpy-cat dataset, (d) AnimalFace-cat dataset and (e) AnimalFace-dog dataset. The decreasing value of FID in red color demonstrates the improvement of InsGen + ATop-k compared with baseline InsGen. *Best viewed in color.*

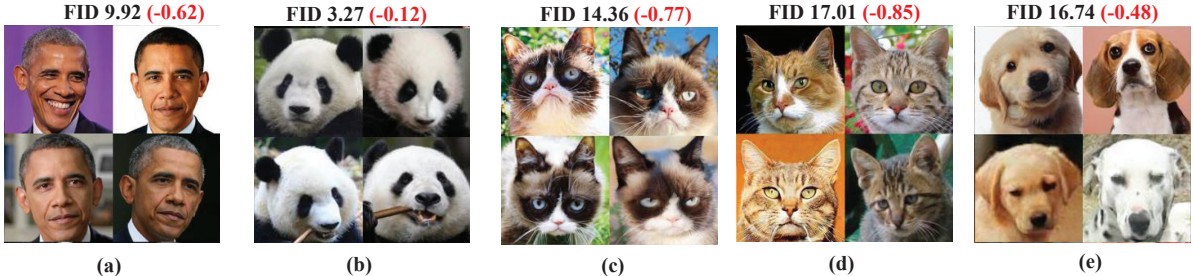

Figure 8: Images generated by Diffusion-Projected GAN + ATop-k on (a) 100-shot Obama dataset, (b) 100-shot Panda dataset, (c) 100-shot Grumpy-cat dataset, (d) AnimalFace-cat dataset and (e) AnimalFace-dog dataset. The decreasing value of FID in red color demonstrates the improvement of Diffusion-Projected GAN + ATop-k compared with baseline Diffusion-Projected GAN.

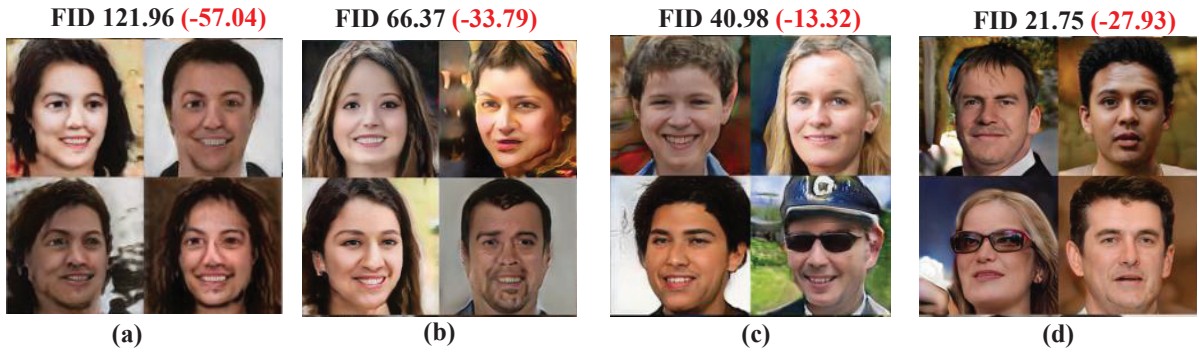

Figure 9: Images generated by StyleGAN2 + ATop-k on (a) FFHQ-100 dataset, (b) FFHQ-1K dataset, (c) FFHQ-2K dataset and (d) FFHQ-5K dataset. The decreasing value of FID in red color demonstrates the improvement of StyleGAN2 + ATop-k compared with baseline StyleGAN2. *Best viewed in color.*

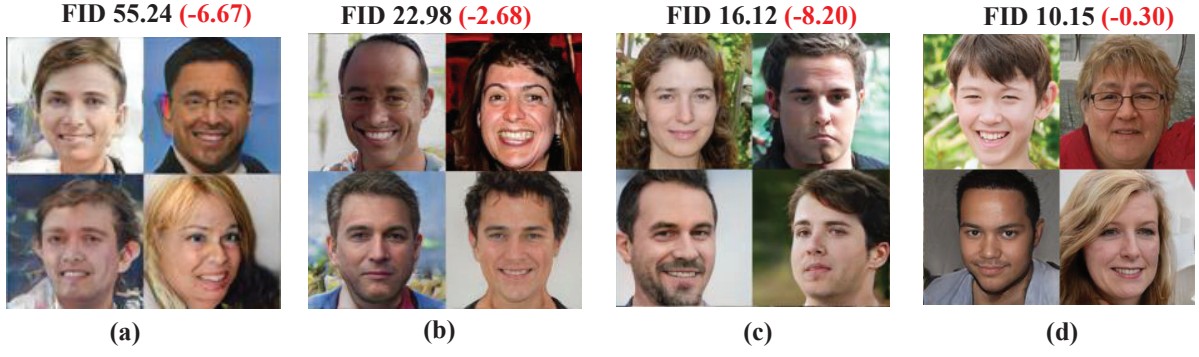

Figure 10: Images generated by StyleGAN2 + Diff-Augment + ATop-k on (a) FFHQ-100 dataset, (b) FFHQ-1K dataset, (c) FFHQ-2K dataset and (d) FFHQ-5K dataset. The decreasing value of FID in red color demonstrates the improvement of StyleGAN2 + Diff-Augment + ATop-k compared with baseline StyleGAN2 + Diff-Augment. *Best viewed in color.*

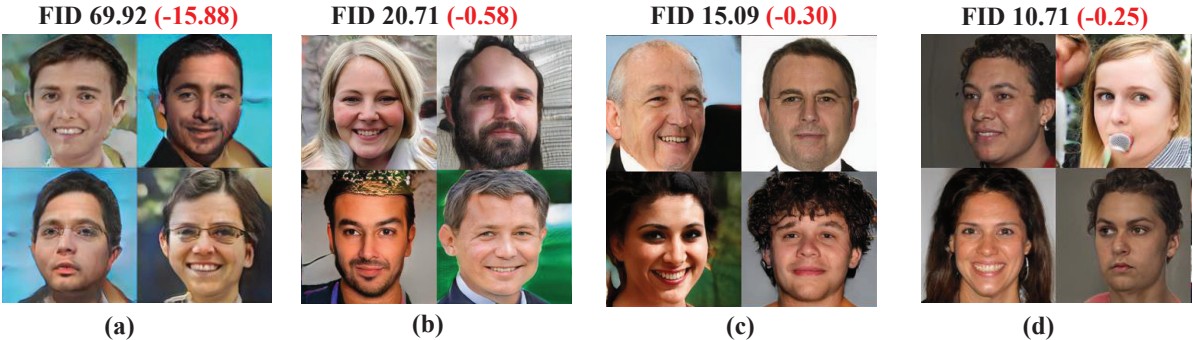

Figure 11: Images generated by StyleGAN2 + ADA + ATop-k on (a) FFHQ-100 dataset, (b) FFHQ-1K dataset, (c) FFHQ-2K dataset and (d) FFHQ-5K dataset. The decreasing value of FID in red color demonstrates the improvement of StyleGAN2 + ADA + ATop-k compared with baseline StyleGAN2 + ADA. *Best viewed in color.*

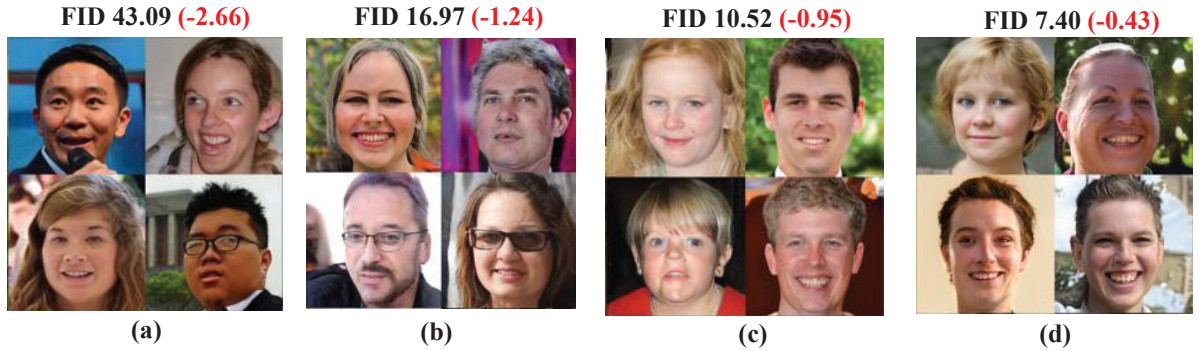

Figure 12: Images generated by InsGen + ATop-k on (a) FFHQ-100 dataset, (b) FFHQ-1K dataset, (c) FFHQ-2K dataset and (d) FFHQ-5K dataset. The decreasing value of FID in red color demonstrates the improvement of InsGen + ATop-k compared with baseline InsGen. *Best viewed in color.*

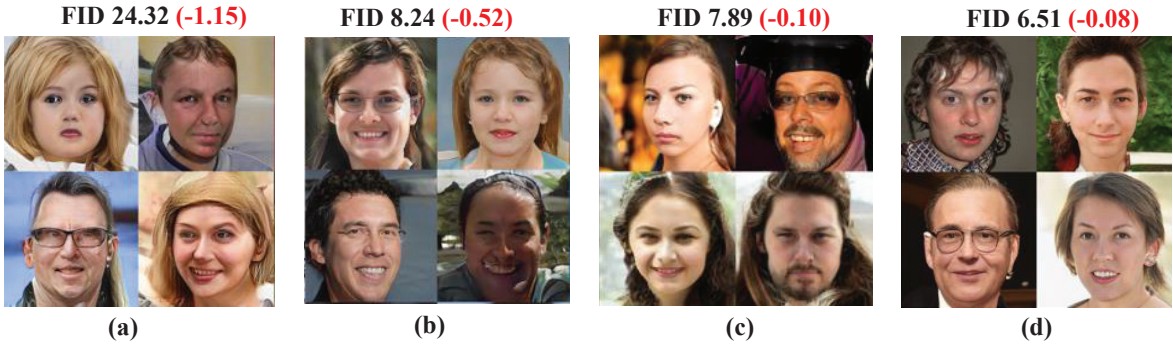

Figure 13: Images generated by Diffusion-Projected GAN + ATop-k on (a) FFHQ-100 dataset, (b) FFHQ-1K dataset, (c) FFHQ-2K dataset and (d) FFHQ-5K dataset. The decreasing value of FID in red color demonstrates the improvement of Diffusion-Projected GAN + ATop-k compared with baseline Diffusion-Projected GAN.

| Method | 100-shot | | | Animal-Face | |
| --- | --- | --- | --- | --- | --- |
| | Obama | Grumpy Cat | Panda | Cat | Dog |
| Diffusion-Projected GAN (Wang et al., 2023b) | 1.67 | 1.47 | 1.00 | 2.28 | 15.32 |
| + ATop-k | **1.68** | **1.48** | **1.01** | **2.35** | **15.61** |

Table 7: The comparison of Inception scores (higher is better) with Diffusion-Projected GAN (Wang et al., 2023b) and Diffusion-Projected GAN + ATop-k on low-shot datasets ($256 \times 256$). We follow the setting as in (Zhao et al., 2020). Massive Augmentation (Cui et al., 2022) is applied to all of the methods. For a fair comparison, the Inception Scores are averaged over three runs; all standard deviations are less than 1% relatively.

| Method | MA | 100% CIFAR-10 | | 20% CIFAR-10 | | 10% CIFAR-10 | |
| --- | --- | --- | --- | --- | --- | --- | --- |
| | | IS | FID | IS | FID | IS | FID |
| StyleGAN2 + Diff-Augment (Zhao et al., 2020) | Yes | 9.40 | 9.89 | 9.21 | 12.15 | 8.84 | 14.50 |
| +ATop-k | Yes | **9.43** | **9.72** | **9.27** | **11.87** | **8.99** | **13.98** |

Table 8: A comparison of the results on the CIFAR-10 dataset (100%, 20% and 10%) with StyleGAN2 + Diff-Augment. Inception Score (IS) and FID are measured using 10k samples; the test set is the reference distribution. For a fair comparison, Massive Augmentation (MA) is applied to all the methods. Results are averaged over five evaluation runs; all standard deviations are less than 1% relatively.

### B.3 More Experiments Results with Inception Score (IS)

To further show the superiority of the proposed ATop-k, we also report the experiment results using another commonly-used GANs evaluation metric, i.e., Inception Score (IS) (Salimans et al., 2016). The results on low-shot datasets compared with the state-of-the-art method, i.e., Diffusion-Projected GAN, are shown in Table 7. Diffusion-Projected GAN + ATop-k can achieve higher IS, demonstrating the superiority of the proposed ATop-k.

### B.4 More Experiments Results on CIFAR-10/100 Datasets

To further demonstrate the superiority of the proposed ATop-k, we also conduct experiments on CIFAR-10/100 datasets compared with StyleGAN2 + Diff-Augment (Zhao et al., 2020). The results are shown in Tables 8 and 9. StyleGAN2 + Diff-Augment + ATop-k achieves better performance on the CIFAR-10/100 datasets.

| Method | MA | 100% CIFAR-100 | | 20% CIFAR-100 | | 10% CIFAR-100 | |
| --- | --- | --- | --- | --- | --- | --- | --- |
| | | IS | FID | IS | FID | IS | FID |
| StyleGAN2 + Diff-Augment (Zhao et al., 2020) | Yes | 10.04 | 15.22 | 9.82 | 16.65 | 9.06 | 20.75 |
| +ATop-k | Yes | **10.09** | **15.06** | **9.91** | **16.37** | **9.22** | **20.02** |

Table 9: A comparison of the results on the CIFAR-100 dataset (100%, 20% and 10%) with StyleGAN2 + Diff-Augment. Inception Score (IS) and FID are measured using 10k samples; the test set is the reference distribution. For a fair comparison, Massive Augmentation (MA) is applied to all the methods. Results are averaged over five evaluation runs; all standard deviations are less than 1% relatively.

| Method | Seconds per 1K images |
|---|---|
| Diffusion-Projected GAN | 12.13 |
| **+ATop-k** | 12.17 |

Table 10: The training time on the 100-shot Obama dataset ($256 \times 256$) with or without ATop-k using Diffusion-Projected GAN backbone. The results are calculated by averaging over ten times on the four NVIDIA RTX TITAN GPUs with batch size 64. All standard deviations are less than 1% relatively.

### B.5 Computational Cost

The results of the training time on the 100-shot Obama dataset with or without ATop-k using Diffusion-Projected GAN have been demonstrated in Table 10. The proposed ATop-k only results in an negligible increase in computational cost.

### B.6 The value of $k$ evolves during training

To demonstrate how the value of $k$ evolves during training, we present its value over a small subset of iterations (50 iterations) for both the early (at 4000 iterations) and late stages (at 50,000 iterations) of the training process. These experiments were conducted on the 100-shot Obama dataset using StyleGAN2 with a batch size of 4, and the results are shown in Figure 14. Furthermore, to further demonstrate the impact of $k$ on the overall training process, in every 4k fake samples shown to $D$, we present the average number of fake samples selected by ATop-k for updating $G$ during training. The results in Figure 15 show that applying Atop-k can select fewer and fewer fake samples to update $G$ during training.

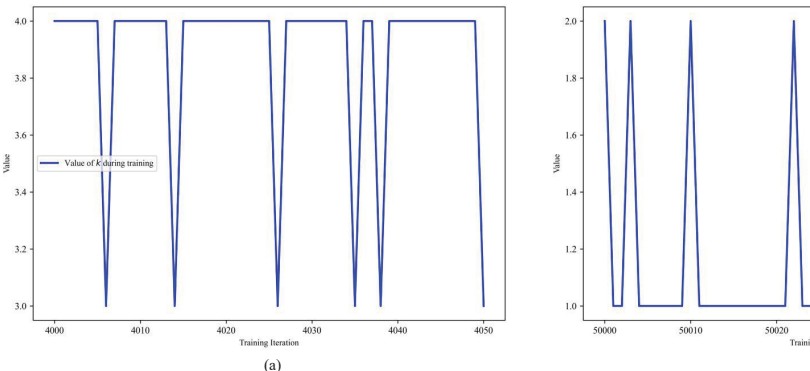

Figure 14: The curves of the values of $k$ during a small subset of iterations. (a) The values of $k$ in the early training stage, from iterations 4000 to 4050. (b) The values of $k$ in the late training stage, from iterations 50,000 to 50,050.

## C   Discussion of Datasets in Experiments

This paper applies the 100-shot Obama dataset, i.e., the dataset consists of Obama faces, in the experiment section. This dataset is widely and commonly used without limitations in DE-GANs research. Furthermore, a lot of recent studies (Zhao et al., 2020; Cui et al., 2022; Chen et al., 2021; Li et al., 2022a; Zhang et al., 2024) on DE-GANs have applied this dataset in their experiments, demonstrating its application is reasonable and does not raise any ethical issues.

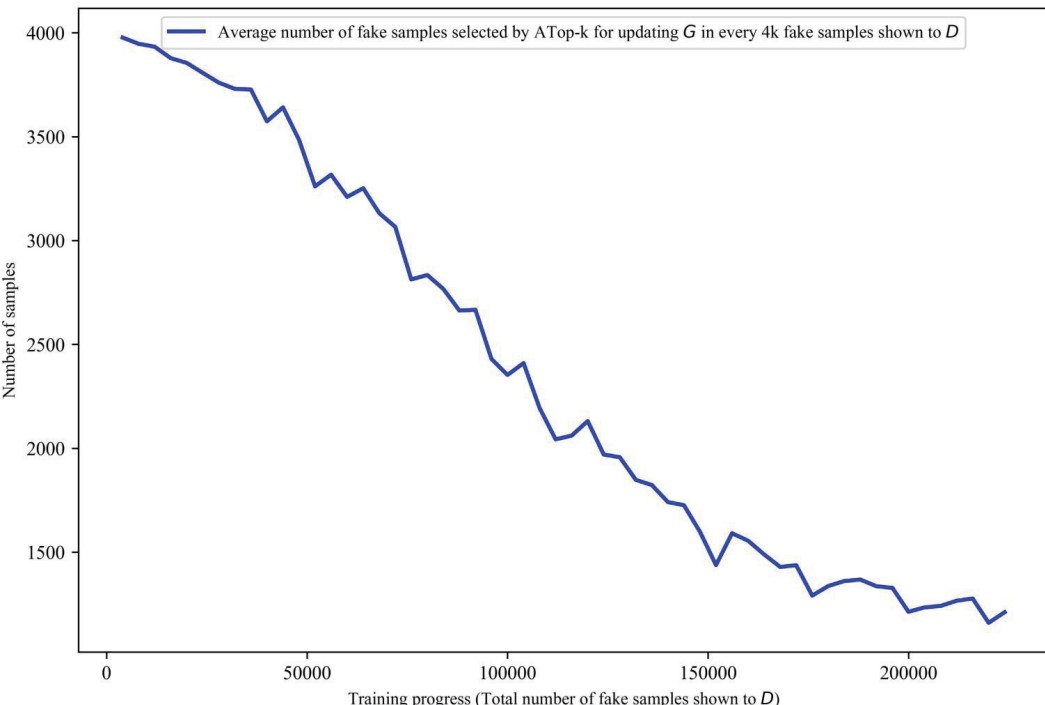

Figure 15: The curves of an average number of fake samples selected by ATop-k for updating $G$ in every 4k fake samples shown to $D$ during training. The experiments are conducted on the 100-shot obama dataset using StyleGAN2 with batchsize 4.

## D    Limitation and Broader Impact

This paper proposes a novel insight of DE-GANs theoretically and seeks to inform future studies on DE-GANs. Driven by this, we propose a simple yet effective method called adaptive Top-k (ATop-k) that can benefit the practical deployment of DE-GANs. The technical contributions of this paper do not raise any particular ethical challenges. However, because technology is usually a double-edged sword, our work may also bring potential social risks when applying GANs with limited data. For example, it may make it easier to generate fake media using only limited data.

