# OpenReview forum: "Understanding and Improving the Training of Data-Efficient GANs"
_TMLR — Rejected by TMLR_

### Review · Reviewer_5tT9 · 2024-10-07

**Summary Of Contributions:**

The contributions of this work are as follows:
- The authors give a novel insight that in DE-GAN training, as the overfitting degree of discriminator D increases, the overlapping parts of real and fake data will become smaller, which makes the useful gradient of generator G smaller and results in training stability and gradient vanishing issues.
- Based on this insight, the authors propose ATop-k strategy to select high-score fake samples while k is adaptive to the overfitting degree of D, which is suggested to provide more useful gradient to G, and authors conducted extensive experiments to validate the claim.

**Audience:**

Yes

**Broader Impact Concerns:**

There should be no potential ethical concerns in this work.

**Claims And Evidence:**

Yes

**Requested Changes:**

- For section 3.1, seems this issue i.e. the disappearance of overlapping part of real and fake data is already pointed out in (1). It would be better if the authors can talk about how data insufficiency is involved in the theory arguments.


(1) Martin Arjovsky and L´eon Bottou. Towards principled methods for training
generative adversarial networks. In International Conference on Learning Representations, 2017.

**Strengths And Weaknesses:**

Strengths
- The main novel point is to pick top k samples to provide more useful gradient to update G. Also this is validated empirically, to illustrate the benefits of selecting k adaptively, the authors conduct experiments on section 4.4 to compare effects of fixed k, linear k, and Atop-k strategy

Weakness
- See requested change

---

> ### Author Response · Authors · 2024-10-24
>
> Dear Reviewer 5tT9,
>
> Thank you for your insightful comments.
>
> **Requested Changes 1. Talk about how data insufficiency is involved in the theory arguments.**
>
> The connection between data insufficiency and the theory in [1] is shown as follows. According to [1], when $D$ reaches the optimal $D^*$, the overlapping parts disappear or can be ignored. Based on Eq.(2) and Eq.(4) in the revised main paper, one key factor for determining the optimal $D^*$ is $P_R$. If $P_R$ consists of sufficient data, $D$ will require a longer time to learn $P_R$ and reach optimal $D^*$. In contrast, when $P_R$ has insufficient data, $D$ can learn $P_R$ and reach $D^*$ much easier, which causes overlapping parts to disappear or be ignored faster.
>
> **Reference**
>
> [1] Martin Arjovsky and L\'eon Bottou. Towards principled methods for training generative adversarial networks. In ICLR 2017.

---

### Review · Reviewer_VzFG · 2024-10-09

**Summary Of Contributions:**

This paper aims to solve the overfitting of the discriminator problem in the data-efficient generative adversarial networks (DE-GANs). The authors mainly propose an adaptive Top-k (ATop-k) method for DE-GANs, which can obtain more useful gradients for G. The ablation studies and visualization results verify its effectiveness. The experiments are conducted on two benchmarks (low-shot and FFHQ) and five baselines (StyleGAN2, StyleGAN2 + Diff-Augment, StyleGAN2 + ADA, InsGen + ADA and DiffusionProjected GAN).

**Audience:**

Yes

**Claims And Evidence:**

No

**Requested Changes:**

Please refer to the above Weaknesses.

1. The paper didn't introduce non-saturating loss.

2. Feature visualization is needed to show that augmenting can overlap the two distributions (i.e., the right sub-figure in Figure 1).

3. Please show how $k$ changes during training.

4. Typo in Sec 4.1: FasetGAN -> FastGAN

**Strengths And Weaknesses:**

**Strengths:**
1. The proposed ATop-k method is easy and effective. It controls the $k$ by converting the logits of D into probabilities using a sign function.
2. The writing and presentation are good.

**Weaknesses:**
1. The connection between analysis (Sec 3.1 & 3.2) and method (Sec 3.3) does not seem strong.

(1) What is the connection between DA and the proposed ATop-k? I don't seem to see it.

(2) Why did you only analyze FastGAN in Sec 3.1?

2. The provided Lemma and Proof provided are too simple. For example, I don't think Proof for Lemma 3 makes sense, and the authors also don't analyze the role of DA when $P_G$ and $P_R$ don't overlap.

3. The very important gradient visualization part is missing, such as the useful gradients of G.


**Questions:**

1. I remember restarting D can alleviate the overfitting problem, does this work for DE-GANs?

2. In Eqn. (8), how to get $p$?

---

> ### Author Response · Authors · 2024-10-24
> **Response Part 1**
>
> Dear Reviewer VzFG,
>
> Thank you for your insightful feedback.
>
> **Weakness 1. The connection between analysis (Sec 3.1 \& 3.2) and method (Sec 3.3) does not seem strong.**
>
> In Sec 3.1, we present a novel theoretical insight in DE-GANs training, regarding the significance of useful gradients of $G$. Then, in Sec 3.2, we apply proposed insight to theoretically illustrate why existing commonly-used approaches, i.e., data augmentation (DA), can improve the DE-GANs training. Driven by this novel insight, in Sec 3.3, we propose a new method called ATop-k to further improve the DE-GANs training. Therefore, Sec 3.1, 3.2 and 3.3 consist of reasonable logic connections.
>
> **Weakness 1 (1). Connection between DA and the proposed ATop-k.**
>
> Based on our response in Weakness 1, both the analysis of DA in Sec 3.2 and the proposed ATop-k in Sec 3.3 aim to demonstrate the significance and generalization ability of the proposed novel theoretical insight in Sec 3.1.
>
> **Weakness 1 (2). Why did you only analyze FastGAN in Sec 3.1.**
>
> We have analyzed both StyleGAN2 with non-saturating loss and FastGAN with hinge loss in Sec 3.1. As pointed out in Reviewer GsVx's comments, we analyze the StyleGAN2 with non-saturating loss by referencing prior methods' theory in [1]. Furthermore, we have added a more detailed analysis of the StyleGAN2 with non-saturating loss in the revised version (shown on page 4 in red color).
>
> **Weakness 2. I don't think Proof for Lemma 3 makes sense, and the authors also don't analyze the role of DA when PG and PR don't overlap.**
>
> Based on empirical results in several existing approaches [2-4], applying DA in GANs has been shown to increase the overlapping parts between real and fake data distributions. The Lemma 3 in the main paper is intended to provide a straightforward theoretical explanation for this observed phenomenon, which is reasonable and meaningful. Furthermore, according to the perfect discrimination theorem in [1], when $D$ reaches optimal $D^*$, the overlapping parts between $P_G$ and $P_R$ disappear or can be ignored. The disappearance of overlapping parts can cause gradient instability or vanishing problems for $G$, which means that $G$ can not be further optimized when $P_G$ and $P_R$ do not overlap. Therefore, existing approaches [2-4] only analyze the effectiveness of their methods when $P_G$ and $P_R$ consist of overlapping parts. We followed these methods and proposed the Lemma 3 in the main paper.
>
> **Weakness 3. The very important gradient visualization part is missing, such as the useful gradients of G.**
>
> As we have already stated on page 1 of the main paper, the concept of useful gradients of $G$ has been explored in existing studies [5-6]. These methods have demonstrated that gradients computed on generated samples that are closer to the real data distribution tend to be more useful for updating the $G$. Following the concept of useful gradients of $G$, visualising the gradients can not effectively demonstrate the usefulness of the gradients. Instead, ADA [2] proposed that the difference between the discriminator's output on real and fake samples can effectively evaluate how close the generator's distribution aligns with the real data distribution (shown in Figure 1 (b) (c) and Figure 6 (b) in ADA paper). Following this conclusion in ADA [2], we have conducted experiments and compared different methods using the difference between discriminator outputs on real and fake samples. The results are shown in Figure 3 (a), which demonstrates that incorporating the proposed ATop-k method in existing methods leads to more useful gradients for $G$.

---

> ### Author Response · Authors · 2024-10-24
> **Response Part 2**
>
> **Question 1. I remember restarting D can alleviate the overfitting problem, does this work for DE-GANs?**
>
> Several existing methods related to restarting $D$ focus on freezing some layers in $D$ [9], restarting the learning rates of $D$ [7] or adjusting the $D$'s capacity [8], which indeed can help address the overfitting of $D$ issue in DE-GANs to some extent. These methods focus on $D$, similar to existing approaches [2-4]. In contrast, our paper presents a novel insight regarding the significance of useful gradients of $G$. Driven by this novel insight, we propose a new method ATop-k, which can directly provide more useful gradients for $G$ during DE-GANs training. Therefore, these restarting $D$ methods can be applied to DE-GANs but they are not directly related to our paper.
>
> **Question 2. In Eqn. (8), how to get p?**
>
> We have already mentioned on page 7 of the main paper (shown on page 6 in the revised paper) that we follow the ADA, APA and ANDA to use $\eta$ to obtain the overfitting probability $p$. We have added more details about this in the revised version (shown on pages 6 in red color).
>
> **Requested Changes 1. The paper didn't introduce non-saturating loss.**
>
> Please refer to our response to Weakness 1 (2) for the details.
>
> **Requested Changes 2. Feature visualization is needed to show that augmenting can overlap the two distributions.**
>
> The empirical results in different existing studies [2-4] have already demonstrated that applying data augmentation (DA) to DE-GANs can increase the overlapping parts between the real and fake data distributions. All of these studies apply the difference between discriminator output on real and fake samples to visualize that augmenting can increase the overlapping parts between $P_G$ and $P_R$. In this paper, we mainly focus on providing a theoretical explanation for the conclusions of these studies. Therefore, the visualization part is not necessary in our paper.
>
> **Requested Changes 3. Please show how k changes during training**
>
> In our paper, the value of $k$ in the proposed ATop-k is determined by the overfitting probability $p$ based on Eq.(9) in the main paper (Eq.(11) in the revised paper). During the training process, the value of the $k$ should exhibit a randomly fluctuating sawtooth pattern between batchsize (early training process when $p$ is small) to 1 (later training process when $p$ is large). However, directly demonstrating how the value of $k$ evolves over a long training process, i.e., across 300k real images shown to $D$ (about 75K iterations with batchsize 4) is infeasible. Therefore, we present the value of $k$ over a small subset of iterations for both the early and late stages of training. These results are detailed in Appendix B.6 of the revised version. Furthermore, to demonstrate how the value of $k$ affects the overall training process, we also present the average number of fake samples selected by ATop-k for updating the generator $G$ for every 4k fake images during training. More details can be found in Appendix B.6 of the revised version.
>
> **Requested Changes 4. Typo in Sec 4.1: FasetGAN -> FastGAN**
>
> Thank you for your careful checking. We have fixed this error in the revised version.

---

> ### Author Response · Authors · 2024-10-24
> **Reference**
>
> **Reference**
>
> [1] Martin Arjovsky and L\'eon Bottou. Towards principled methods for training generative adversarial networks. In ICLR 2017.
>
> [2] Tero Karras, Miika Aittala, Janne Hellsten, Samuli Laine, Jaakko Lehtinen, and Timo Aila. Training generative adversarial networks with limited data. In NeurIPS 2020.
>
> [3] Liming Jiang, Bo Dai, Wayne Wu, and Chen Change Loy. Deceive d: Adaptive pseudo augmentation for GAN training with limited data. In NeurIPS 2021.
>
> [4] Zhengli Zhao, Zizhao Zhang, Ting Chen, Sameer Singh, and Han Zhang. Image augmentations for gan training. arXiv preprint arXiv:2006.02595, 2020.
>
> [5] Samarth Sinha, Zhengli Zhao, Anirudh Goyal ALIAS PARTH GOYAL, Colin A Raffel, and Augustus Odena. Top-k training of GANs: Improving GAN performance by throwing away bad samples. In NeurIPS 2020.
>
> [6] Yan Wu, Jeff Donahue, David Balduzzi, Karen Simonyan, and Timothy Lillicrap. LoGAN: Latent optimisation for generative adversarial networks. arXiv preprint arXiv:1912.00953, 2019.
>
>
> [7] Radford, Alec. Unsupervised representation learning with deep convolutional generative adversarial networks. arXiv preprint arXiv:1511.06434 (2015).
>
> [8] Yang, C., Shen, Y., Xu, Y., Zhao, D., Dai, B., & Zhou, B. Improving gans with a dynamic discriminator. NeurIPS 2022.
>
> [9] Mo, Sangwoo, Minsu Cho, and Jinwoo Shin. Freeze the discriminator: a simple baseline for fine-tuning gans. arXiv preprint arXiv:2002.10964 (2020).

---

### Review · Reviewer_GsVx · 2024-10-11

**Summary Of Contributions:**

The paper tackles the problem of vanishing gradient in GAN training. Specifically, it proposes an adaptive Top-K selection heuristic to select the best fake examples, because these samples can better help the generator learn the real data distribution. The paper also points out the insight that disjoint support of the real and generated data can lead to instability or a vanishing gradient problem.

**Audience:**

Yes

**Broader Impact Concerns:**

The paper does not have ethical implications.

**Claims And Evidence:**

Yes

**Requested Changes:**

* The authors need to present the related theorem in the paper so it's self-contained.

* Can the authors answer my concern regarding the theory?

**Strengths And Weaknesses:**

* The paper makes frequent references to prior works' theory, which are not present in the paper. The writing is not self-contained.

* The method lacks novelty: the proposed method seems to be a minor update from the existing top-K selection for GAN training [1] regarding methodology and motivation.

* I am particularly concerned about the proof of Lemma 2. The lemma uses Thereom 2.1 and 2.2 from [2], which states that if the supports of real and generated data are disjoint, the perfect discriminator will be $0$ on the support of the generated data.  This is true for a discriminator $D: x \rightarrow [0,1]$. However, the discriminator in hinge loss [3] $D(x) := <w, \phi(x)>+b$ , which is not necessarily within $[0,1]$.

[1] Sinha, Samarth, et al. "Top-k training of gans: Improving gan performance by throwing away bad samples." Advances in Neural Information Processing Systems 33 (2020): 14638-14649.

[2] Arjovsky, Martin, and Léon Bottou. "Towards principled methods for training generative adversarial networks." arXiv preprint arXiv:1701.04862 (2017).

[3] Lim, Jae Hyun, and Jong Chul Ye. "Geometric gan." arXiv preprint arXiv:1705.02894 (2017).

---

> ### Author Response · Authors · 2024-10-24
>
> Dear Reviewer GsVx,
>
> Thank you for your insightful comments.
>
> **Weakness 1 and Requested Changes 1. The paper makes frequent references to prior works' theory, which are not present in the paper. The writing is not self-contained.**
>
> The theories presented in [1] are about 7 pages long. Directly rephrasing all of them in our paper is infeasible. Therefore, in the previous version, we opted to cite the specific lemma and theorem in [1] to maintain the conciseness and clarity of our paper. In the revised version, to further improve the self-sufficiency of our paper, we have presented the name of the theories rather than the specific lemma and theorem and briefly extended the theory of StyleGAN2 with non-saturating loss, shown on page 4 in red color.
>
> **Weakness 2. The method lacks novelty.**
>
> Our paper contributes both a theoretical insight regarding the significance of useful gradients of $G$ and a new method called adaptive Top-k. Specifically, we first unveil a novel insight in DE-GANs training, regarding the significance of useful gradients of $G$ (Sec 3.1). Based on this, we propose a novel method called adaptive Top-k. The difference between the proposed ATop-k and the existing Top-k method [2] is shown in Sec 2.3. In summary, we believe both our theoretical insights and proposed method will inform future research on DE-GANs.
>
> **Weakness 3 and Requested Changes 2. Can the authors answer my concern regarding the theory?**
>
> The main purpose of Lemma 2 in the main paper is to demonstrate that GANs with hinge loss suffer from the gradient vanishing of $G$ under the optimal discriminator $D^*$, where the optimal $D^*$ is the sign function between the $P_G$ and $P_R$ shown in Lemma 1 in the main paper. It is clear that for $\forall x \in (-\infty, +\infty)$, if the $P_G$ and $P_R$ consist of the support contained on two disjoint compact subsets, $D^*(x)$ has accuracy 1
> and $\bigtriangledown_xD^*(x)=0$ (this is because the $\bigtriangledown_x\text{sign}(x)\equiv 0$). In this case, the perfect discrimination theorems in [1] is still satisfied under $D^* : \mathcal{X} \to (-\infty,+\infty)$. We have improved the Lemma2 accordingly in the revised paper.
>
> **Reference**
>
> [1] Martin Arjovsky and L\'eon Bottou. Towards principled methods for training generative adversarial networks. In ICLR 2017.
>
> [2] Sinha, Samarth, et al. Top-k training of gans: Improving gan performance by throwing away bad samples. In NeurIPS 2020.

---

### Decision · Action_Editor_h15R · 2024-11-20

**Recommendation:** Reject

**Comment:**

Overall, the consensus from reviewers was to reject this paper. The authors provided a meaningful rebuttal, but the reviewers did not feel that their concerns were fully addressed. I would suggest the authors consider resubmission after addressing the reviewer's comments and the concerns highlighted under "claims & evidence".

**Audience:**

Yes.

**Claims And Evidence:**

Reviewers generally raised a few concerns about the clarity of the paper:
- There were concerns about the validity of some of the theoretical arguments made in the paper, as raised by multiple reviewers.
- One reviewer had trouble connecting the paper's theory to the proposed algorithm.
- Reviewers also had questions about how the theory connects, makes use of, and builds on the work of Arjovsky and Bottou. and had suggestions about clarifying/summarizing this past work.

**Resubmission Of Major Revision:**

The authors may consider submitting a major revision at a later time.